# Dynamic Plasma Lipidomic Analysis Revealed Cholesterol Ester and Amides Associated with Sepsis Development in Critically Ill Patients after Cardiovascular Surgery with Cardiopulmonary Bypass

**DOI:** 10.3390/jpm12111838

**Published:** 2022-11-03

**Authors:** Wenyan Ding, Shaohang Xu, Baojin Zhou, Ruo Zhou, Peng Liu, Xiangyi Hui, Yun Long, Longxiang Su

**Affiliations:** 1Department of Critical Care Medicine, Peking Union Medical College Hospital, Peking Union Medical College, Chinese Academy of Medical Sciences, Beijing 100730, China; 2Deepxomics Co., Ltd., Shenzhen 518000, China; 3Medical Research Center, Peking Union Medical College Hospital, Chinese Academy of Medical Sciences and Peking Union Medical College, Beijing 100730, China; 4State Key Laboratory of Complex Severe and Rare Diseases, Peking Union Medical College Hospital, Peking Union Medical College, Chinese Academy of Medical Sciences, Beijing 100730, China

**Keywords:** lipidomics, cardiac surgery, sepsis, machine learning, signature

## Abstract

Background: Sepsis in patients after cardiovascular surgery with cardiopulmonary bypass (CPB) has a high rate of mortality. We sought to determine whether changes in lipidomics can predict sepsis after cardiac surgery. Methods: We used high-performance liquid chromatography coupled to tandem mass spectrometry to explore global lipidome changes in samples from a prospective case-control cohort (30 sepsis vs. 30 nonsepsis) hospitalized with cardiovascular surgery. All patients were sampled before and within 48–72 h after surgery. A bioinformatic pipeline was applied to acquire reliable features and MS/MS-driven identifications. Furthermore, a multiple-step machine learning framework was performed for signature discovery and performance evaluation. Results: Compared with preoperative samples, 94 features were upregulated and 282 features were downregulated in the postoperative samples of the sepsis group, and 73 features were upregulated and 265 features were downregulated in the postoperative samples of the nonsepsis group. “Autophagy”, “pathogenic Escherichia coli infection” and “glycosylphosphatidylinositol-anchor biosynthesis” pathways were significantly enriched in the pathway enrichment analysis. A multistep machine learning framework further confirmed that two cholesterol esters, CE (18:0) and CE (16:0), were significantly decreased in the sepsis group (*p* < 0.05). In addition, oleamide and stearamide were increased significantly in the postoperative sepsis group (*p* < 0.001). Conclusions: This study revealed characteristic lipidomic changes in the plasma of septic patients before and after cardiac surgery with CPB. We discovered two cholesterol esters and two amides from peripheral blood that could be promising signatures for sepsis within a dynamic detection between the preoperative and postoperative groups.

## 1. Introduction

Systemic inflammatory response syndrome (SIRS) and sepsis occur frequently after cardiac surgery with cardiopulmonary bypass (CPB). The mortality rate of sepsis varies from 17% to 79%, as previously reported [1]. Contact with artificial surfaces, myocardial and pulmonary ischemia/reperfusion, and surgery were considered to be the main causes of postoperative SIRS [2], while sepsis with suspected or proven infection was diagnosed in 4.7% and 4.8% of patients, respectively [3]. It is difficult to distinguish noninfective SIRS from sepsis after surgery. Delayed diagnosis will directly affect the prognosis of patients.

The mechanism of sepsis in patients after cardiopulmonary bypass is still unclear. It was reported that most of the early bloodstream infections for patients undergoing CPB were caused by Gram-negative bacteria [4], while prolonged CPB could increase intestinal permeability and lead to endotoxin or bacterial translocation from the intestine to the bloodstream [5]. Previous studies have revealed that disorders of the intestinal flora and metabolites may cause systemic inflammatory reactions and eventually lead to sepsis [6]. The gut microbiota has been shown to affect lipid metabolism and lipid levels in blood and tissues in humans [7]. Moreover, in terms of carriers of cholesterol, intravenous application of reconstituted HDL reduced inflammation in both plasma and organs in endotoxemic rodent models [8], and low cholesterol levels before cardiac surgery with CPB are a potential signature to evaluate the risk of sepsis [9,10], indicating that sepsis after cardiac surgery may be closely related to lipid metabolism. Furthermore, lipid peroxidation increases the concentrations of superoxide (O_2_^−^) and peroxynitrite (ONOO^−^), which are frequently linked to poor clinical outcomes, including in-hospital death, development of postoperative myocardial infarction and postoperative atrial fibrillation [11,12,13,14]. In addition, lipid metabolites such as circulating ketones and β-hydroxybutyrate could protect patients against heart failure by reducing inflammation [15]. Therefore, it is interesting to explore the global lipid signatures associated with sepsis after cardiac surgery.

In this study, we performed plasma lipidomic analysis of patients after cardiac surgery with extracorporeal circulation to explore sepsis-related lipid signatures. We conducted bioinformatic analyses and machine learning methods to explore the biological significance of novel lipidomic markers in patients with sepsis and to illustrate the mechanism of the inflammatory response after cardiac surgery with CPB.

## 2. Materials and Methods

### 2.1. Study Design Patients and Setting

This study was designed as a prospective observational case-control study according to the Strengthening the Reporting of Observational Studies in Epidemiology (STROBE) guidelines. Patients who underwent sternotomy and CPB due to cardiac surgery was performed by the Department of Cardiac Surgery of Peking Union Medical College Hospital. The protocol was ethically approved by the ethics committee of Peking Union Medical College (ID: ZS-1612) and filed at ClinicalTrials.gov (NCT04032938); accessed on 29 July 2019. All participants gave informed consent. The study was performed in accordance with the Declaration of Helsinki. Patients admitted to the intensive care unit (ICU) after cardiac surgery with CPB were divided into two groups based on their clinical diagnosis: the sepsis group and the nonsepsis Group 48–72 h after surgery. After fasting for at least 6 h but not more than 10 h, venous blood was collected (without anticoagulation) by direct venepuncture or established venous access from enrolled patients. Plasma was taken from the blood and aliquoted in 4 Eppendorf tubes to remain at −80 °C for 1 h. Blood samples were obtained 24 h before surgery and 48–72 h after surgery. Sepsis was defined when the patients were treated with systemic therapeutic administration of antibiotics due to suspected or proven infection and with an increase in SOFA score by 2 or more points [16]. Suspected infection could be defined as the preventive administration of oral or parenteral antibiotics and sampling of body fluid cultures (blood, urine, cerebrospinal fluid, peritoneal, etc.). Patients with proven infection were identified as those who had body fluids sampled for culture with pathogenic organisms. Patients were excluded if they (1) had a fever before surgery, regardless of the etiological evidence of infection; (2) had anti-infective treatment before surgery; (3) had a history of CPB in 6 months; or (4) rejected or abandoned ICU therapeutic intervention. Sample collection was terminated when both groups received 30 cases. The flowchart is shown in Figure 1. Finally, a total of 60 consecutive patients were enrolled.

### 2.2. Study Procedures and Blood Sampling

Detailed baseline and clinical information were recorded from the subject/proxy and bedside assessments were performed using medical records. Patient identification information was removed and replaced with sample numbers. Blood samples were drawn into pyrogen-free tubes, and plasma was separated by centrifugation into Eppendorf tubes. Samples were stored frozen at −80 °C until subsequent analyses.

### 2.3. Sample Preparation and Lipid Extraction

Prior to the experiment, samples were left at −20 °C for 30 min and then thawed at 4 °C until no ice was observed in the tubes. The lipid extraction method followed a previously published paper [17]. Briefly, 40 μL of plasma and 1 μL of stable isotopic lipid standards (Avanti Polar Lipids, Alabaster, AL, USA) were extracted with 120 μL precooled isopropanol (IPA) and then vortexed for 1 min. After incubation for 10 min at room temperature, the mixture was stored overnight in a refrigerator at −20 °C to improve protein precipitation. Samples were centrifuged for 20 min at 14,000× *g*, and then the supernatant was further diluted with IPA/acetonitrile (ACN)/H2O (2:1:1 *v*:*v*:*v*) and stored at −80 °C until LC–MS analysis. Equal amounts of all samples were pooled as a QC sample for LC–MS system conditioning and quality control [18].

### 2.4. Lipid Detection with UPLC–MS/MS

The extracted lipids were separated on an Ultimate 3000 RSLC system (Thermo Fisher Scientific, Waltham, MA, USA) with an ACQUITY UPLC CSH C18 column (2.1 × 100 mm, 1.7 μm, Waters, Milford, MA, USA) and emitted into a Q-Exactive mass spectrometer (Thermo Fisher Scientific, Waltham, MA, USA). Mobile phase A consisted of 10 mM ammonium formate and 0.1% formic acid (ACN:H2O = 60:40, *v*/*v*), and mobile phase B consisted of 10 mM ammonium formate and 0.1% formic acid (IPA:ACN = 90:10, *v*/*v*). A flow rate of 0.4 mL/min was used. The initial elution was started at 40% B and was immediately increased by a linear gradient to 43% B for the first 2 min, followed by an increase to 50% B within 0.1 min. Over the next 3.9 min, the gradient was increased to 54% B, and the amount of B was increased to 70% during the next 0.1 min. In the final part of the gradient, B was increased to 99% and maintained for 1.9 min. Finally, B was returned to 40% over the next 0.1 min and equilibrated for 1.9 min for the next injection. To enhance lipid detection coverage, the Q-Exactive was set to both positive and negative modes in a top-3 configuration to acquire data in DDA mode.

### 2.5. Data Processing

LC−MS raw data files were converted into mzXML format and then processed by the XCMS [19] and metaX [20] toolboxes implemented with R language. The acquired MS data pretreatments, including peak picking, peak grouping, retention time correction and second peak grouping, were performed using the XCMS package. Each ion was identified by combining retention time (RT) and m/z data. The intensities of each peak were recorded, and a three-dimensional matrix containing arbitrarily assigned peak indices (retention time-m/z pairs), sample names (observations) and ion intensity information (variables) was generated. The intensity of the peak data was further processed by metaX. Those features that were detected in less than 50% of QC samples or 80% of biological samples were removed, and the remaining peaks with missing values were imputed with the k-nearest neighbor algorithm and normalized using the probabilistic quotient normalization method. Quality control-based robust LOESS signal correction was fitted to the QC data with respect to the order of injection to minimize signal intensity drift over time [18]. In addition, the relative standard deviations of the metabolic features were calculated across all QC samples, and those >50% were then removed.

### 2.6. Lipid Identification

Lipids were identified with fragment spectrum matching using MSDIAL 2.94 software(RIKEN Center for Sustainable Resource Science, Yokohama City, Kanagawa, Japan) against a combination MS2 database (Lipidblast, MetDNA, Massbank, HMDB). We set mass tolerance to 0.01 for MS1 and 0.05 for MS2. The identification score cut-off was set to 0.75.

### 2.7. Statistical Analysis

All statistical analyses were performed using the R platform for statistical computing and graphics (R version 4.0.2, Revolution Analytics, Mountain View, CA, USA). Univariate analysis and multivariate analysis were performed by metaX. PCA was performed to detect outliers, and PLSDA was applied using log transformation and Pareto scaling. Permutation testing (200 times) on the R2 and Q2 of the PLSDA was used to assess the reliability of the PLS-DA model [21]. The unpaired Wilcoxon test was performed to test significant differences between the control and experimental groups, and the *p* value was adjusted for multiple hypothesis testing using the Benjamini–Hochberg method. Differential lipids were obtained using the Wilcoxon test with the criteria of fold change (FC) ≥1.5 and adjusted *p* value ≤ 0.05.

### 2.8. Sepsis Related-Signature Discovery

The same filtered and normalized lipid abundance matrix used in the differential expression analysis was also used as input for signature discovery and performance evaluation. The candidate lipids were first selected by a two-way ANOVA model with an FDR cut-off ≤0.05 between sepsis and nonsepsis samples. The significant lipids are reported in Appendix A, and the corresponding hierarchical clustering-based heatmap was generated by the “pheatmap” R package. Next, we used a Monte Carlo cross validation (MCCV) framework in MetaboAnalyst [22], coupled with a random forest algorithm, to construct the discriminative model. Two-thirds of the samples are used in each MCCV for determining feature importance. The top 2, 3, 5… Forty-three (max) important features are separately utilized to construct corresponding classification models. Specifically, each classification tree in RF is developed by a random feature selection from a bootstrap sample at each branch. The majority vote of the ensemble is used to determine the class prediction. Other relevant information provided by RF include OOB (out-of-bag) error and variable importance measure. Approximately one-third of the samples are left out of the bootstrap instance during tree construction. This OOB data is then utilized as a test sample to calculate an unbiased estimation of the classification error (OOB error). A variable is considered important in permutation analysis if it has a favorable impact on prediction accuracy, estimated by the OOB prediction error. The ROC curve and radar chart were created by ROCR [23] and ggradar packages with the R environment, respectively.

## 3. Results

### 3.1. Patient Characteristics

All relevant clinical and monitoring data were prospectively collected from 371 adult patients admitted to the Department of Critical Care Medicine after cardiac surgery at Peking Union Medical College Hospital of China between August 2019 and September 2020. A total of 60 patients were enrolled based on exclusion criteria. Among them, 30 patients were diagnosed with sepsis (as the case group), and 30 were diagnosed with nonsepsis (Figure 1). Complete patient (*n* = 30:30) characteristics for the study population are shown in Table 1. There was a tendency towards higher levels of APACHE II and PCT in the sepsis group. The duration of mechanical ventilation time and ICU stay in the sepsis group were significantly longer than those in the nonsepsis group (*p* = 0.042, *p* < 0.001).

### 3.2. Case Characteristics

The Sepsis-3 criteria for sepsis were met by 30 patients. Sepsis with suspected infection occurred in 18 patients (60%), and sepsis with proven infection was demonstrated in 12 (40%). Patients in the suspected infection group were identified as those who had body fluids sampled for culture and received antibiotics but had a negative culture result. Patients in the documented infection group were identified as those who had body fluids sampled for culture with a positive result. The respiratory tract was the most frequent source of both proven (75%) and suspected infection (72.2%). Other sources of infection are shown in Table 2.

### 3.3. Data Quality Assessment

The untargeted lipidomic analysis yielded 25,883 features in positive ion mode (PIM) and 25,734 features in negative ion mode (NIM). The majority of lipid features were located in the 100–1200 m/z range (Figure 2A,B). The different distributions of lipids in positive and negative modes indicated that there was good complementarity in the two detection modes (Figure 2C,D). Principal component analysis (PCA) showed that all of the QC samples spiked at certain intervals clustered together, verifying acceptable reproducibility and stability of the results (Figure 2E,F). In addition to QC samples, we also evaluated the reproducibility of seven stable isotope lipid standards spiked into samples before lipid extraction with extracted ion chromatogram (Appendix A), peak area (Appendix A) and retention time (Appendix A). The trend of seven internal standards further highlighted the high quality of the dataset, which paved the way for further statistical analyses.

For the lipid identifications, there were 966 (positive mode) and 673 (negative mode) features involved in more than 23 lipid subclasses annotated by accurate precursor mass and MS/MS matching with the spectral library. The summary and detailed identification information can be acquired in Appendix A and Appendix A, respectively. The retention time distribution shows that lipids of different classes can be well separated according to hydrophobicity (Appendix A). Furthermore, we explored the difference in retention time between lipids in our experiment and the lipids in the Lipidblast library. The larger positive correlation coefficients provided a higher confidence of relative identification (Appendix A).

### 3.4. Differential Lipid Analysis for Surgery

To obtain the differentially expressed lipids, all detected features were evaluated using the Wilcoxon test with the criteria of fold change (FC) ≥1.5 and adjusted *p* value ≤ 0.05. Detailed statistical information, including FC and adjusted *p* value for each feature, can be found in Appendix A. Then, the lipid features with significant differences before and after surgery are displayed in two volcano plots in Figure 3A,B. Specifically, 94 features were upregulated and 282 features were downregulated in the SPost group (sepsis patients after surgery) compared with their corresponding samples before surgery (SPre group). For the NPost (control patients after surgery) group compared with the NPre (control patients before surgery) group, 73 features were upregulated and 265 features were downregulated. Moreover, we used an UpSet plot to visualize the set intersections of different groups. Remarkably, there were 39 features that were significantly increased in only septic patients (Figure 3C). These features may have the potential to be important diagnostic markers. To further explore the biological functions of the detected features, we performed Kyoto Encyclopedia of Genes and Genomes (KEGG) pathway enrichment analysis to annotate the potential functional implications (Appendix A, Figure 3D). The ‘‘autophagy’’ and “glycosylphosphatidylinositol-anchor biosynthesis” pathways were highly enriched in these differential features, suggesting that they play a crucial role in the repair or recovery process after surgery. “Pathogenic Escherichia coli infection” entry indicated that clinical syndromes might be associated with infection by pathogenic *E. coli* strains.

### 3.5. Signature Discovery and Evaluation Related to Sepsis

We used our lipid datasets to perform the discovery and evaluation of potential signature molecules related to sepsis. By comparing sepsis and nonsepsis samples after surgery with the Wilcoxon test, we found very few significantly differential lipids. Therefore, we inferred that in a clinical lipidomic dataset with surgical intervention effects, solely relying on a nonparametric statistical hypothesis test for signature discovery may be too rigid. Thus, we introduced a multistep machine learning workflow to help to explore potential signatures (Figure 4A). We first used a two-way ANOVA method to obtain 43 significantly differentially expressed features between SPost and NPost samples (Appendix A). Among these features, we found that most of the sphingolipids and fatty acyls were highly expressed in the SPost samples, while glycerolipids and sterol lipids were significantly upregulated in NPost samples (Figure 4B). Then, an MCCV framework coupled with a random forest algorithm was used to establish this assessment model. As shown in Figure 4C, we found that each model had a high area under the ROC curve. The top 10 features based on the selected frequency are displayed in Figure 4D.

In general, simple models with a smaller number of signatures are more robust and less prone to overfitting. Therefore, we chose the top five significant lipids, ranked based on their selected frequencies, as the optimal cut-off of the signature. Specifically, two cholesterol esters, CE (18:0) and CE (16:0), were significantly decreased in the sepsis group *(p =* 0.005, *p* < 0.001). In addition, oleamide and stearamide, the top two lipids, were two amides derived from the fatty acid oleic acid and stearic acid, respectively. These two lipids were increased significantly in the postoperative sepsis group (*p <* 0.001). Both of them had a higher abundance in sepsis samples, indicating that these markers may have a potential relationship with sepsis. Moreover, 6,10,14-trimethyl-5,9,13-pentadecatrien-2-one, also named farnesyl acetone, which has a role in hormonal regulation of metabolism, was upregulated in sepsis cohorts. For example, farnesyl pyrophosphate (FPP), as a ligand for the glucocorticoid receptor (GR), can cause activation of the glucocorticoid (GC) signaling pathway in primary human epidermal keratinocytes, thereby exerting hormone regulation effects [24]. To ensure the credibility of the five signatures, we provided the corresponding extracted ion chromatogram (EIC) and high-quality MS/MS annotated spectra in Appendix A. To explore the expression change of these specific lipids, we used a box-whisker plot to illustrate the alteration of three fatty acids or three amides and five cholesterol esters from the chemical structure and evolution (Figure 5).

## 4. Discussion

Sepsis is one of the critical postoperative complications after cardiac surgery [25]. However, the underlying mechanisms contributing to sepsis after cardiovascular surgery with CPB are still not clear. Previous metabolomic studies and meta-analyses on sepsis suggested that phospholipids were significantly enriched in several death-related metabolic pathways [26]. However, studies on lipidomic analyses are limited, and this is the first study to investigate lipid alterations and their potential molecular mechanisms for sepsis after cardiovascular surgery with CPB by UPLC–MS/MS.

Comparing preoperative and postoperative samples, there were 94 upregulated and 282 downregulated features in the SPost (sepsis patients after surgery) group and 73 upregulated and 265 downregulated features in the NPost (nonsepsis patients after surgery) group. Autophagy, pathogenic Escherichia coli infection and glycosylphosphatidylinositol-anchor biosynthesis were significantly enriched in the postsurgery groups. It has been reported that autophagy is closely related to the innate immune system, which can alleviate excessive inflammatory responses and act as a key factor for sepsis development [27,28]. Cardiac autophagy in sepsis is recognized as a cellular adaptive protective mechanism by limiting cell damage and apoptosis [29]. The previous experimental results of our team also showed that mitochondrial uncoupling protein 2 (UCP2) may play a protective role against LPS by regulating the balance between autophagy and apoptosis of cardiomyocytes [30]. The “Pathogenic *Escherichia coli* infection” entry indicated that there may be infection of the pathogenic *E. coli* strain. In the present study, two patients with sepsis had a positive result of *Escherichia coli* in sputum culture. A previous report pointed out that cardiopulmonary bypass might cause hypoperfusion of the small intestinal mucosa and consequently bacterial translocation [31]. Our research provides a basis for this view, and there is an ongoing project from our team on gut microbiota and the mechanism of sepsis after CPB [32]. Glycosylphosphatidylinositol (GPI)-anchored proteins are numerous on the surface of eukaryotic cells and participate in a wide variety of physiological functions, such as the complement cascade, the pro- and anti-inflammatory responses of macrophages, and the activation, development, and proliferation of T cells [33,34,35,36]. Our study revealed that the glycosylphosphatidylinositol-anchor biosynthesis pathway was activated in postoperative specimens, which is in accordance with a previous study showing that the downregulation of GPI-PLD could play an important role in proinflammatory responses [37].

In this study, we showed that sepsis cases can be distinguished by molecular signatures using a machine learning framework based on the expression levels of 5 metabolites. This model achieved an area under the curve (AUC) of 0.963 in the training set. As shown in the KEGG map pathway, palmitic acid can be elongated to stearic acid, and stearic acid will be further desaturated to produce oleic acid [38]. Due to the significantly elevated abundances of fatty acids and relevant fatty acid amides in patients undergoing surgery (Figure 5A), it is hypothesized that inflammatory or stress responses to surgical trauma or wound repair may lead to increased expression of some lipids. A previous study demonstrated that fatty acids can improve the inflammatory, proliferative, and remodeling phases of wound healing by inducing angiogenesis at the wound site [39]. Therefore, our data provide further evidence that the systemic inflammatory cascade to infection will lead to striking increases in palmitamide, stearamide and oleamide. The characteristic molecular changes of these three fatty acid amides have important potential applications in building a diagnostic model for identifying sepsis cases. Oleamide and stearamide were the top two upregulated lipids in the septic group after surgery. Oleamide plays a protective role in sepsis. Specifically, connexin43 (Cx43), a member of a large family of transmembrane proteins, plays a vital role in the process of cell damage amplification and deterioration in sepsis [40]. Oleamide is an effective inhibitor of Cx43 channels [41], inhibiting ROS transfer and inactivating the JNK1/Sirt1/FoxO3a signaling pathway, thereby preventing damage caused by sepsis.

Cholesterol esters of CE (18:0) and CE (16:0) were significantly decreased in the sepsis group in our study. Several previous findings confirmed that there was a crucial connection between cholesterol metabolism and innate immunity [42,43,44]. Recent research from Hu et al. demonstrated that serum cholesterol is remarkably decreased among Chinese patients with COVID-19 [45]. Another report suggested that cholesteryl ester transfer protein had the ability to disrupt the interplay of bacterial lipopolysaccharide and TLR4 (Toll-like receptor 4), thereby reducing the uncontrolled inflammatory response in sepsis [46]. Medications such as statins can help to lower cholesterol levels. However, statins have no effect on mortality outcomes in patients with sepsis compared with placebo [9]. Some studies proved that low HDL cholesterol levels correlate with the severity of sepsis during infection [47], and patients with low levels of LDL cholesterol (LDL-C) have an increased risk of sepsis and worse outcomes [48]. Nevertheless, there is no sufficient evidence to confirm the direct relationship between CE and sepsis. Herein, two cholesterol esters (CE16:0 and CE18:0) from five signatures were significantly decreased in comparison to SPost/NPost after surgery. Interestingly, we also obtained similar observations from the comparison of SPre/NPre before surgery. We further explored more abundances of CEs from the candidate features of our data, and all the boxplots exhibited similar trends in Figure 5B. This evidence provides sufficient reason to speculate that people with low cholesterol have weak immune systems and are prone to bacterial infections, which will eventually trigger sepsis after surgery. We proposed that CE can act as an effective predictor for the early detection of septic patients before surgery.

Nevertheless, some limitations of our study were as follows. First, although sepsis patients have a higher mortality rate after cardiac surgery combined with CPB, no deaths occurred in the patients we included. The differences may be more pronounced if we introduced critical patients in the sepsis cohort. Second, all patients received prophylactic use of second-generation cephalosporins during the surgery, which may affect the manifestation of infection. Third, the 5 significant lipids were not validated in a larger cohort to predict sepsis in clinical practice or animal experiments. Further studies on gut microbiota and lipid metabolism in sepsis from clinical or animal studies will be implemented to verify these signatures.

## 5. Conclusions

Overall, UPLC–MS/MS-based lipidomic profiles revealed that two cholesterol esters (CE16:0 and CE18:0) were significantly decreased in the sepsis group and nonsepsis group both before and after surgery. This work suggests that CE could serve as a promising predictor for septic patients before surgery. Oleamide and stearamide were the top two significantly upregulated lipids that exhibited excellent performance in distinguishing sepsis after cardiac surgery with CPB. Therefore, we identified a potential intervention site at which regulating cholesterol and lecithin cholesterol acyltransferase might inhibit the progression of sepsis in the early stage.

## Figures and Tables

**Figure 1 jpm-12-01838-f001:**
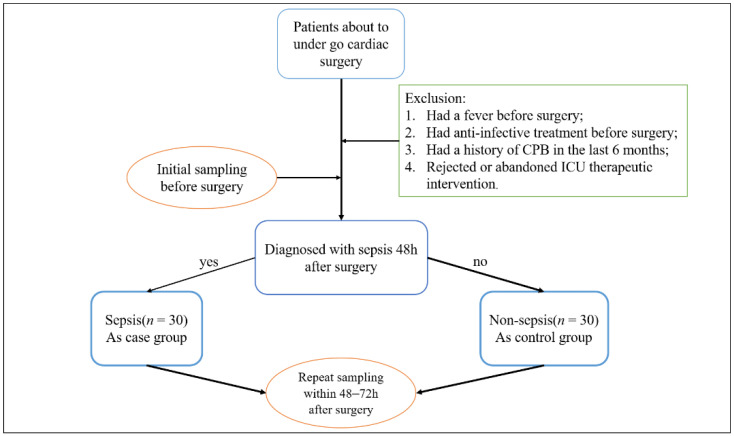
Clinical trial flowchart.

**Figure 2 jpm-12-01838-f002:**
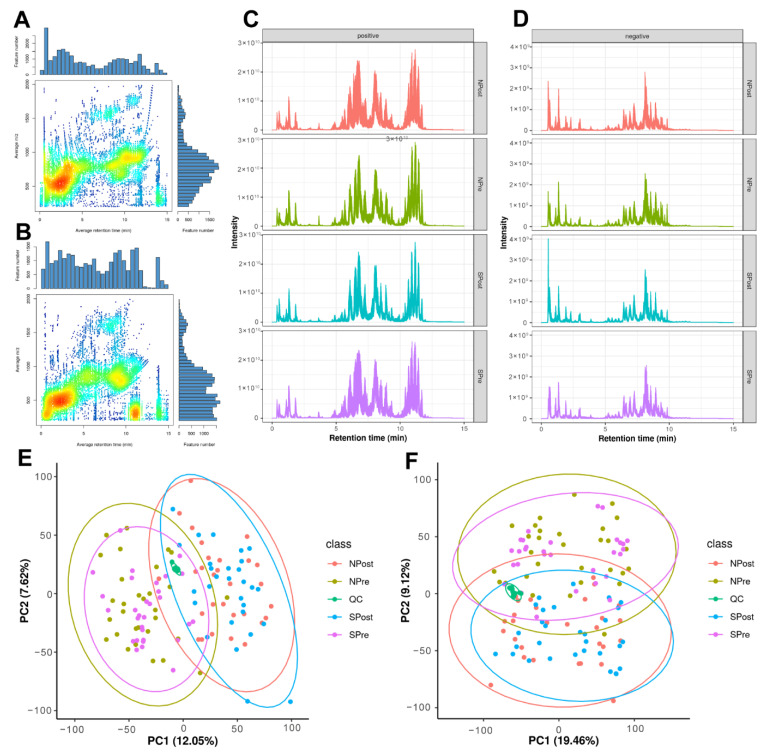
Overview of quantitative lipidomic data. The aligned feature distribution in positive mode (**A**) and in negative mode (**B**). Total ion chromatogram in positive mode (**C**) and in negative mode (**D**). Principal component analysis of all lipids in positive mode (**E**) and in negative mode (**F**).

**Figure 3 jpm-12-01838-f003:**
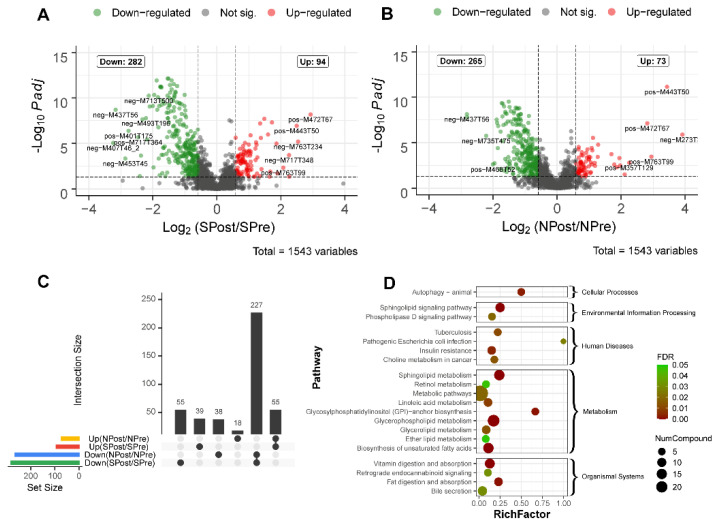
Differential lipids for sepsis patients (*n* = 30) before and after surgery. Volcano plot of sepsis (*n* = 30) (**A**) and nonsepsis patients (*n* = 30) (**B**). The x-axis is the mean ratio fold-change (plotted with a log 2 scale) of the relative abundance of each feature between the patients after and before surgery. The y-axis represents the adjusted *p* value (FDR) of the ratio fold-change for each feature. An UpSet diagram shows differentially expressed identifications that are unique or shared among four comparison groups (**C**). Pathway enrichment analysis of differentially expressed lipids is presented in the bubble chart (**D**). The y-axis corresponds to KEGG pathway items, and the x-axis represents the enrichment factor. The color of the dot represents FDR, and a lower FDR indicates more significant enrichment. The point size represents the number of differentially expressed lipids that are mapped to the reference pathways. The rich factor refers to the level of enrichment, which is the ratio of the differentially expressed lipids annotated in this pathway to all lipids annotated in this pathway item. A greater enrichment factor indicates a more significant enrichment.

**Figure 4 jpm-12-01838-f004:**
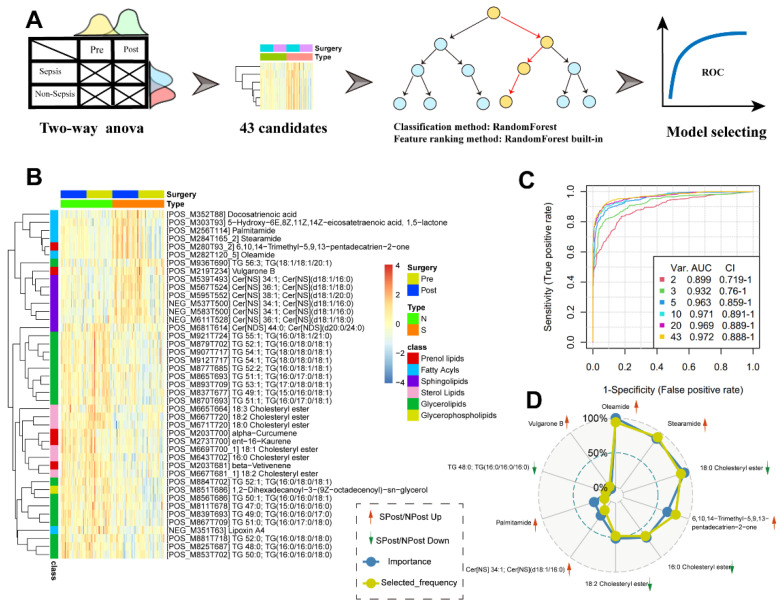
(**A**) A flow chart to demonstrate the main steps of signature discovery and evaluation. (**B**) Forty-three candidate signatures from two-way ANOVA. (**C**) Performance evaluation of top 2, 3, 5 … 43 (max) important features to construct corresponding classification models. (**D**) Radar plots of the top 10 features with the highest selection frequency in the classification models.

**Figure 5 jpm-12-01838-f005:**
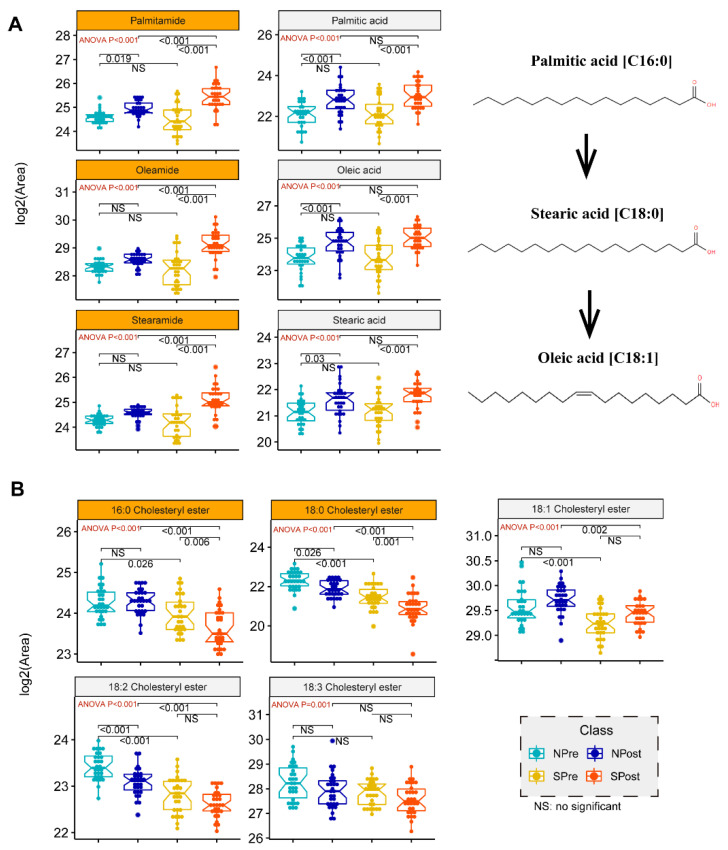
Box plot analysis for candidate signatures of amides or fatty acids. ANOVA with posthoc tests were used to determine the *p* values of pairwise comparison between groups. The corresponding *p* values are labelled in the top area of the figure. The red texts in the upper left corner are the *p* values of ANOVA tests, and the black texts in the top are the *p* values of Tukey post-hoc tests. The orange in the facet background indicates that this feature is one of the candidate signatures from the multistep machine learning workflow. (**A**) Schematic of palmitic, stearic and oleic acid biosynthetic pathways derived from KEGG pathway maps. (**B**) Box-and-whisker plot for the general trends of 5 cholesteryl esters.

**Table 1 jpm-12-01838-t001:** Clinical characteristics of the subjects.

Variable	Sepsis	Nonsepsis	*p* Value
Age, mean (range), year.	52.6 (27–73)	51.7 (24–80)	0.806
Female sex, *n* (%)	15 (50)	16 (53.3)	0.796
Height, mean (SD), cm	163.7 (8.4)	164.7 (8.4)	0.635
Weight, mean (SD), kg	66.5 (11.7)	65.9 (12.8)	0.870
BMI, mean (SD), kg/m^2^	24.7 (3.3)	24.2 (3.6)	0.554
APACHE II (when entering the ICU)	13.7(5.5)	11.1 (4.9)	0.065
SOFA (48–72 h after surgery)	6.7 (3.3)	5.9 (3.6)	0.351
Duration of CPB, median (interquartile range), min	139.5 (96–175.5)	117.5 (79.5–154.3)	0.157
Requirement for renal replacement therapy, *n* (%)	2 (6.7)	2 (6.7)	-
Duration of mechanical ventilation (range), h	93.9 (15–272)	58.3(5–259)	0.042
ICU duration, days	9.4	5.5	*<*0.001
PCT, ng/mL	16	6.3	0.057

SOFA, Sequential Organ Failure Assessment, APACHE II, Acute Physiology and Chronic Health Evaluation, PCT, procalcitonin. The sepsis group and nonsepsis group were compared using the Wilcoxon signed rank test. Proportions were compared using Chi square.

**Table 2 jpm-12-01838-t002:** Suspected or proven sources of infection in those diagnosed with sepsis.

Suspected Source	Suspected Infection	Proven Infection
Respiratory	13	9
Abdominal/gastrointestinal	0	1
Wound	2	0
Genitourinary	0	0
Bacteremia/catheters	3	2
CSF	0	–
Dental	0	–

CSF, cerebrospinal fluid.

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
