# Peer review of "Dynamic Plasma Lipidomic Analysis Revealed Cholesterol Ester and Amides Associated with Sepsis Development in Critically Ill Patients after Cardiovascular Surgery with Cardiopulmonary Bypass"

_jpm, 2022, doi:10.3390/jpm12111838_

Round 1

Reviewer 1 Report

Original article by Wen Yan Ding et al. entitled ‘Dynamic Plasma Lipidomic Analysis Revealed Cholesterol Ester and Amides Associated with Sepsis Development in Critically Ill Patients after Cardiovascular Surgery with cardiopulmonary bypass’ deal with the important topic of sepsis following CPB surgery. Initially, the study is looking nice and authors found some interesting data related to the development of sepsis and non-sepsis group following surgery. Some of the questions raised after careful review of this manuscript.

There is no strong relation or data available related to this topic. Authors should clarify about that. I think the most recent publication in 2007.

References used in this study are not recent. Please add some recent findings related to this topic to strength your study.

Plasma have all the coagulating factors including albumin whether authors used any technique to remove those factors before lipidomic assay? What if samples contaminated with those factors and what would be the effect on analysis? While the limitations of this study are already discussed by the authors in the manuscript.

Whether authors performed lipid profile of those patients by analyzer?

Whether authors have any data related to fat percentage in those subjects? BMI?

Author Response

1. There is no strong relation or data available related to this topic. Authors should clarify about that. I think the most recent publication in 2007.

Response: Thank you for your advice to emphasize that this is the first study to investigate lipid alterations and their potential molecular mechanisms for sepsis after cardiovascular surgery with CPB by UPLC–MS/MS. We added it in discussion part.

2. References used in this study are not recent. Please add some recent findings related to this topic to strength your study.

Response: Thank you for your suggestions to update our references. we added ref7,8,9 to the manuscript.

3. Plasma have all the coagulating factors including albumin whether authors used any technique to remove those factors before lipidomic assay? What if samples contaminated with those factors and what would be the effect on analysis? While the limitations of this study are already discussed by the authors in the manuscript.

Response: Thank you for this question. We agree that coagulating factors before lipid analysis should be considered. In fact, our laboratory has evaluated these factors systematically in the technique development majoring in plasma lipidomic analysis based on qualitative and quantitative criteria, such as protein removal efficiency, selectivity, repeatability, and recovery efficiency. These evaluation results were consistent with previous reported, as cold isopropanol can effectively precipitate proteins with the removal of condensation factors, resulting in high recovery and good reproducibility for lipid analysis. [https://pubs.acs.org/doi/abs/10.1021/ac500317c]. In this study, we further used seven lipid standards added into each plasma sample before plasma lipid extraction. As shown in Figure S1B, the good reproducibility of internal standards revealing the interference of coagulation factors during sample preparation was limited and would not affect lipid analysis.

4. Whether authors performed lipid profile of those patients by analyzer?

Response: Thanks for asking. Actually, we only did postoperative lipids profile for patients, and there was no difference in postoperative Triglycerides between the two groups(P=0.462).

5. Whether authors have any data related to fat percentage in those subjects? BMI?

Response: Thanks for your good advice. We supplemented the BMI statistics in Table 1, showing no statistical difference.

Reviewer 2 Report

The purpose of this study is to determine whether changes in lipidomics can predict sepsis after cardiac surgery. The study is designed as a prospective case-control cohort study in which patients were divided into two groups based on their clinical diagnosis: the sepsis group and the non-sepsis group 48-72 hours after surgery. This work suggests that two cholesterol esters may serve as promising predictors for septic patients before surgery. According to the study, oleamide and stearamide were the two major lipids that showed excellent performance in discriminating sepsis after cardiac surgery with CPB.

General comments:

The sample size of both study groups is 30+30 patients, which is a rather small sample size due to the demanding statistical and machine learning methods used in lipodomic analysis and further statistical analyses.

Data analyses 

Lipid extraction, detection, and identification involve machine learning methods (e.g., PCA, k-means clustering, random forest classification) that are highly dependent on sample size and open parameter tunning. The training process was according to the paper done on two-thirds and validated on one-third of the samples. Since the sample size is small, it would be better to do one of the cross validation techniques or better leave-one-cross validation. Please explain the training and validation process in a little more detail.

Statistical analysis

Did you use nonparametric statistics in all statistical analyses (e.g., Wilcoxon test)? Are all data non-normally distributed? Please explain the use of non-parametric tests in all your statistical analyses.

In results section you stated that simple models with a smaller number of signatures are more robust and less prone to overfitting. Therefore, we chose the top 5 significant lipids, ranked based on their selected frequencies, as the optimal cut-off of the signature.

Please explain why you selected 5 significant lipids. According to what criteria?

Box plot analysis in Figure 5: Please elaborate the usage of pair-wise Wilcoxon tests with P-value corrections instead of ANOVA with posthoc analysis.  

Minor Comment:

Figure 5: Change scientific notation to normal notation (e.g 6.6e-8 should be better written to < 0.001)

Reviewer 3 Report

By this paper, authors try to find out a possibile lipidic signature characterizing patients with sepsis after cardiac surgery.

Topic is nicely presented and resulting data are interesting, promising and pretty solid; overall this paper is good however, let me point out some minor changes/corrections which can improve this manuscript.

- Paragraph 2.4

While describing UPLC-MS-MS method, both mobile phase A and B consist of 10 mM ammonium formate and 0.1% formic acid. Please check it.

- Paragraph 3.1

Table 1 row “Weight mean SD” column “Sepsis” data is reported as (11.) as it seems a number is missing. Please check it

- Paragraph 3.3

In the text is missing the reference to figure 2C and 2 D. Please check it.

Author Response

1.While describing UPLC-MS-MS method, both mobile phase A and B consist of 10 mM ammonium formate and 0.1% formic acid. Please check it.

Response:

Thank you for your question. We confirmed that the description of the mobile phase was correct. The additive composition of the mobile phase was followed a previously published paper [https://www.nature.com/articles/s41587-020-0531-2].

2.Table 1 row “Weight mean SD” column “Sepsis” data is reported as (11.) as it seems a number is missing. Please check it.

Response: Thank you for your careful review. The number should be 11.7. We revised it in the manuscript.

3. In the text is missing the reference to figure 2C and 2 D. Please check it.

Response: Thank you for your careful review. We added the figure 2C and 2 D in the manuscript(Line 218).

Round 2

Reviewer 1 Report

Authors have addressed most of the comments.

In abstract line 33..We discovered two cholesterol esters and two amides..

I would use 'found' instead of discovered...because the concept of this research is in very early phase even the hypothesis is not established yet (lipid levels and their role in sepsis).

My first comment was about the objective of the study, and the answer is still not convincing. I would suggest authors to increase sample size to get more reproducible data.